# HETEROLOGOUS NORMALIZATION

## ABSTRACT

Batch Normalization has become a standard technique for training modern deep networks. However, its effectiveness diminishes when the batch size becomes smaller since the batch statistics estimation becomes inaccurate. This paper proposes Heterologous Normalization, which computes normalization's mean and standard deviation from different pixel sets to take advantage of different normalization methods. Specifically, it calculates the mean like Batch Normalization to maintain the advantage of Batch Normalization. Meanwhile, it enlarges the number of pixels from which the standard deviation is calculated, thus alleviating the problem caused by the small batch size. Experiments show that Heterologous Normalization surpasses or achieves comparable performance to existing homologous methods, with large or small batch sizes on various datasets.

## 1 INTRODUCTION

Deep neural networks have received great success in many areas. Batch Normalization (BN) has become a standard technique for training modern deep networks. BN normalizes the features by the mean and the standard deviation (std) computed from a batch of samples. The coordination between examples helps the learning process. The random selection of examples in the minibatch brings the sampling noises, providing a regularization effect. However, its effectiveness diminishes when the batch size becomes smaller since the noises are too much and make inaccurate batch statistics estimation. That hinders BN's usage for scenes lacking samples or memories. For example, federated learning, a hot topic in machine learning, aims to train a model across multiple decentralized edge devices or servers to address privacy and security issues. The heterogeneous environments require a robust training algorithm with large or small batch size.

To address the small batch size problem, some methods try to avoid normalizing along the batch dimension. In the case of NCHW format feature map, let N refer to the batch dimension, C refer to the channel dimension, H and W refer to the spatial height and width dimensions. Layer Normalization (LN) (Ba et al., 2016) computes the mean and the std along (C,H,W) dimensions. Instance Normalization (IN) (Ulyanov et al., 2016) computes the mean and the std along (H,W) dimensions. Group Normalization (GN) (Wu & He, 2018) is an intermediate state between Layer Normalization and Instance Normalization. It uses a group of channels within the sample itself to compute the mean and the std. Avoiding normalizing along the batch dimension gives up the advantages of BN, resulting in poor performances in many cases. To take advantage of different approaches, Switchable Normalization (SN) (Luo et al., 2018a) combines different normalizations. It uses learnable parameters to combine BN, IN, and LN linearly. SN brings more statistics and computational complexity. Besides, the mean and the std of those methods need to be calculated at inference time. They slow down the inference comparing to BN since BN's mean and std are pre-computed from the training data by the moving average.

Although different normalizations compute the statistics from different pixel sets (pixel refers to a element in the feature map), a specific normalization computes the mean and the std from the same pixel set. Thus existing methods can be viewed as homologous normalization. In this paper, we propose Heterologous Normalization (HN), which computes the mean and the std from different pixel sets to take advantage of different normalization methods. Although HN is a general method that can use different strategies to compute the mean and the std, we find this combination strategy works well in most situations: calculating the mean along the (N, H, W) dimensions as the same as BN, while calculating the std along the (N,C,H,W) dimensions. On the one hand, it maintains the advantage of batch normalization when the batch size is large. On the other hand, it enlarges the

number of pixels from which the std is calculated, thus alleviating the problem caused by the small batch size. At inference time, HN's mean and std are pre-computed from the training set by moving average as the same as BN, keeping the inference efficiency.

We evaluate HN on various datasets: CIFAR-10, CIFAR-100, CalTech256, ImageNet. Experiments show that heterologous combination is valid for most situations. HN surpasses or achieves comparable performance than BN, IN, LN, GN, and SN, with large or small batch sizes on various datasets. Moreover, HN can be combined with SN together to improve the performance further. By analyzing the statistics' evolution over the course of training, we find that the noise of the small batch size is mainly caused by the std fluctuation rather than the mean. Enlarging the number of pixels from which the std is calculated can alleviate fluctuation successfully. That explains why HN works well with the small batch size.

We conclude our key contributions as follows: 1) We show that it is unnecessary to estimate normalization statistics from the same pixel set and propose a general Heterologous Normalization that calculates normalization statistics from different pixel sets. 2) We find a special heterologous method that surpasses or achieves comparable performance to existing homologous methods, with large or small batch sizes on various datasets. 3) We find the noise of the small batch size is mainly caused by the std fluctuation rather than the mean. We should make an equilibrium between generalization and stability by controlling the number of pixels to calculate the statistics.

## 2 RELATED WORK

Batch Normalization (BN) (Ioffe & Szegedy, 2015) is effective at accelerating and improving the training of deep neural networks by reducing internal covariate shift. It performs the normalization for each training minibatch along (N,H,W) dimensions in the case of NCHW format feature. Since BN uses the statistics on minibatch examples, its effect is dependent on the minibatch size. To mitigate this problem, Normalization Propagation (Arpit et al., 2016) uses a data-independent parametric estimate of the mean and standard deviation instead of explicitly calculating from data. Batch Renormalization (Ioffe, 2017) introduces two extra parameters to correct the fact that the minibatch statistics differ from the population ones. It needs to train the model for a certain number of iterations with Batch Normalization alone, without the correction, then ramps up the amount of allowed correction.

There is a family of methods that avoid normalizing along the batch dimension. Layer Normalization (LN) (Ba et al., 2016) computes the mean and standard deviation along (C,H,W) dimensions. Instance Normalization (IN) (Ulyanov et al., 2016) computes the mean and standard deviation along (H,W) dimensions. When the batch size is 1, Batch Normalization is equivalent to Instance Normalization. Group Normalization (GN) (Wu & He, 2018) is an intermediate state between Layer Normalization and Instance Normalization. It uses a group of channels to compute the mean and the std, while Layer Normalization uses all channels, and Instance Normalization uses one channel. To take advantage of different approaches, Switchable Normalization (SN) (Luo et al., 2018a), Exemplar Normalization Zhang et al. (2020) and Batch-Instance Normalization (BIN) (Nam & Kim, 2018) try to combine different normalization together. Switchable and Exemplar Normalization use learnable parameters to combine Batch, Instance, and Layer Normalization. BIN uses a learnable gate parameter to combine Batch and Instance normalization.

Weight normalization (Salimans & Kingma, 2016) normalizes the filter weights instead of the activations by re-parameterizing the incoming weight vector. Cosine normalization (Luo et al., 2017) normalizes both the filter weights and the activations by using cosine similarity instead of dot product in neural networks. Some researchers try to use other statistics instead of the mean and the standard deviation in normalization. Instead of the standard $L^2$ batch normalization, (Hoffer et al., 2018) uses normalization in $L^1$ and $L^\infty$ spaces and shows that it can improve numerical stability in low-precision implementations as well as provide computational and memory benefits. Generalized batch normalization (Yuan et al., 2019) investigates a variety of alternative deviation measures for scaling and alternative mean measures for centering. Virtual batch normalization (Salimans et al., 2016) and spectral normalization (Miyato et al., 2018) focus on the normalization in generative adversarial networks. Self-Normalizing (Klambauer et al., 2017) focuses on standard feed-forward neural networks (fully-connected networks). Recurrent batch normalization (Cooijmans et al., 2016) modifies batch normalization to use in recurrent networks. Kalman normalization (Wang et al.,

2018) estimates the mean and standard deviation of a certain layer by considering the distributions of all its preceding layers, mimicking the merits of Kalman Filtering Process. EvalNorm (Singh & Shrivastava, 2019) estimates corrected normalization statistics to use for batch normalization during evaluation. (Ren et al., 2016) provides a unifying view of the different normalization approaches. (Santurkar et al., 2018), (Luo et al., 2018b) and (Bjorck et al., 2018) try to explain how batch normalization works. Summers & Dinneen (2020) propose several useful techniques to improve batch normalization.

## 3 HETEROLOGOUS NORMALIZATION

We first describe some notations that will be used next. For NCHW format feature map, let $U$ denote the universal pixel set in the same feature layer, $U_N$ denote the set of pixels that belong to the same example, $U_C$ denote the set of pixels that belong to the same channel, $U_G$ denote the set of pixels that belong to the same group of channels. A family of normalization can be formalized as:

$$\widehat{x}_i = \frac{1}{\sigma_{S_i'}}(x_i - \mu_{S_i}) \tag{1}$$

$$y_i = \gamma \widehat{x}_i + \beta \tag{2}$$

where $x_i$ is the input, and $y_i$ is the output of the normalization. $\mu$ is the mean, and $\sigma$ is the standard deviation (std). $S_i$ is the pixel set from which the mean is computed, and $S_i'$ is the pixel set from which the std is computed. $\gamma$ and $\beta$ are learned parameters for affine transformation.

As shown in Table 1, different normalization methods estimate statistics from different pixel sets. BN computes the mean and the std along the (N, H, W) dimensions. The random noise brought by batch statistics estimation is beneficial to the generalization. However, the batch statistics estimation brings too much noise when the batch size is small. IN, LN, and GN try to avoid estimate statistics along the batch dimension. Each example estimates statistics within its own pixel sets, ignoring the information of other examples. Since there are not enough pixels to estimate statistics for BN when batch size is small, a straightforward method is to extend the pixel set for statistics calculation. For example, we can use the universal pixels $U$ or the pixels from a group of channels $U_G$ to calculate statistics. We use Extended Normalization (EN) to refer to that method. In the following, the default number of groups in EN is 1 (the universal set $U$) when there is no mention of it, and we use EN_G$n$ to represent that the groups of channels are $n$ in EN. Figure 1 shows the visualization of different normalizations.

Table 1: Different normalization

|     | $S_i = S_i'$ | Description |
| --- | --- | --- |
| BN | $U_C$ | Estimate statistics from pixels that belong to the same channel. |
| IN | $U_C \cap U_N$ | Estimate statistics from pixels that belong to both the same channel and the same example. |
| LN | $U_N$ | Estimate statistics from pixels that belong to the same example. |
| GN | $U_G \cap U_N$ | Estimate statistics from pixels that belong to both the same group of channels and the same example. |
| EN | $U_G$ | Estimate statistics from pixels that belong to the same group of channels. |

Although different normalizations compute the statistics from different pixel sets, a specific normalization computes the mean and the std from the same pixel set. Thus existing methods can be viewed as homologous normalization, which has

$$S_i = S_i' \tag{3}$$

In this paper, we propose Heterologous Normalization (HN), which computes the mean and the std from different of pixel sets to take advantage of different normalization methods. That is to say, in HN, we have

$$S_i \neq S_i' \tag{4}$$

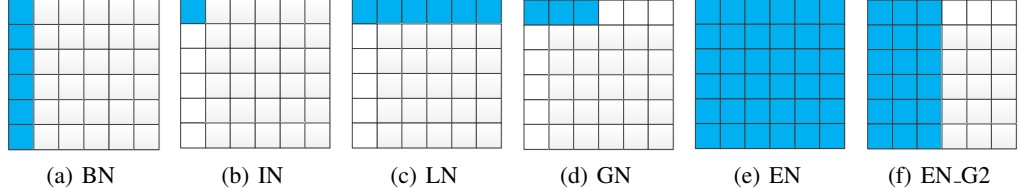

|                |                |                |                |                |                |
| :------------: | :------------: | :------------: | :------------: | :------------: | :------------: |
| (a) BN | (b) IN | (c) LN | (d) GN | (e) EN | (f) EN_G2 |

Figure 1: Visualization of different normalization methods. Horizontal is channel dimension, and vertical is batch dimension. The small square grid is the pixel set along the spatial height and width dimensions.

Although HN is a general method that can use different strategies to compute the mean and the std, we find this configure works well in most situations:

$$S_i = U_C, \; S_i^{'} = U_G \tag{5}$$

Specifically, the mean is calculated along the (N, H, W) dimensions as the same as BN, while the std is computed along the (N,C,H,W) dimensions as the same as EN. On the one hand, it maintains BN's advantage when the batch size is large. On the other hand, it enlarges the number of pixels from which the std is calculated, thus alleviating the problem caused by the small batch size.

Switchable Normalization (SN) (Luo et al., 2018a) also try to take advantage of different normalizations by mixing them together. Its mean and std are both linear combinations of the batch, instance, and layer ones. The pixel sets, from which the mean and the std are calculated, are the same. Different from SN, HN calculates the mean and the std from heterogeneous pixel sets. HN has fewer statistics and computational complexity than SN. Besides, the mean and the std of HN are pre-computed from the training data by the moving average as the same as BN. There is no need to compute the mean and the std at inference time. Moreover, since the mean and the std are pre-computed and fixed at inference time, the normalization can further be fused into convolution operation. That is very helpful to speed up the inference, especially on mobile or embedded devices. As for IN, LN, GN, SN, the mean and the std need to be calculated at inference time. Finally, HN and SN are not mutually exclusive. We can combine HN and SN together. For example, we can use the mean of BN, and the linear combinations of the std from different normalizations.

## 4  EXPERIMENT

### 4.1  CIFAR-10

CIFAR-10 (Krizhevsky & Hinton, 2009) is a data set of natural 32x32 RGB images in 10 classes with 50, 000 images for training and 10, 000 for testing. The training images are padded with 0 to 36x36 and then randomly cropped to 32x32 pixels. Then randomly horizontal flipping is made. Each channel of the input is normalized globally. Weight decay of 0.0005 and SGD with 0.9 momentum are used. For the batch size of 128, the initial learning rate is set to 0.1. For other batch sizes $n$, the initial learning rate is set to $0.1 * n/128$, following the linear scaling rule. We train the networks for 200 epochs and decrease the learning rate by 10x at 100 and 150 epochs. One GPU is used to train all networks.

**Different combination strategies.** Figure 2 show the test accuracy of ResNet18 (He et al., 2016) using different normalization combination. For simplicity, we use (A,B) to represent the mean is calculated as the same as normalization A (column in Figure 2), and the std is calculated as the same as normalization B (row in Figure 2). And we use * to represent any normalization. The diagonal cell is the traditional homologous normalization whose mean and std are calculated from the same pixel set.

When the batch size is 128, as shown in Figure 2(a), (BN,BN) gets better performance than other homologous normalization, e.g. (IN,IN), (LN,LN), (GN,GN), (EN,EN). Heterologous combinations (BN,*) all achieve decent performances except (BN,IN). Among them, (BN,EN) achieves the best accuracy 95.18%, which is higher than (BN,BN) 95.05%. Many other combinations also show heterologous normalization is effective. For example, heterologous (IN,LN) 93.33% and (LN,IN)

93.10% are better than homologous (IN,IN) 92.88% and (LN,LN) 92.21%. When the batch size is 4, as shown in Figure 2(b), the accuracy of (BN,BN) deteriorates seriously, suffering the issues caused by the small batch size. We find that heterologous normalization is generally valid. Most heterologous combinations not only surpass the homologous combinations but also outperform the corresponding ones with large batch size 128. Moreover, heterologous combinations (BN,*) all achieve decent performances. Among them, (BN,EN_G32) achieves the best accuracy 95.04%, comparable to (BN,BN) 95.05% with large batch size 128.

| mean / std | BN | IN | LN | GN | EN | EN_G32 |
|---|---|---|---|---|---|---|
| BN | 95.05 | 94.63 | 93.74 | 93.94 | 93.41 | 94.20 |
| IN | 93.62 | 92.88 | 93.10 | 93.18 | 92.14 | 93.52 |
| LN | 94.44 | 93.33 | 92.21 | 92.63 | 85.87 | / |
| GN | 94.72 | 93.68 | 93.39 | 93.76 | 91.06 | 94.21 |
| EN | 95.18 | 93.51 | 90.76 | 91.60 | 91.66 | 92.04 |
| EN_G32 | 94.97 | 93.76 | 92.40 | 93.24 | 92.31 | 93.52 |

(a) Batch size = 128

| mean / std | BN | IN | LN | GN | EN | EN_G32 |
|---|---|---|---|---|---|---|
| BN | 84.59 | 93.85 | 94.18 | 94.48 | 94.19 | 94.61 |
| IN | 93.42 | 93.47 | 93.22 | 93.70 | 93.24 | 94.02 |
| LN | 94.93 | 93.66 | 92.90 | 93.74 | 93.39 | 93.69 |
| GN | 94.82 | 93.50 | 93.95 | 94.06 | 93.27 | 94.21 |
| EN | 94.93 | 93.62 | 93.62 | 94.37 | 93.69 | 94.37 |
| EN_G32 | 95.04 | 93.93 | 93.76 | 94.42 | 93.70 | 94.17 |

(b) Batch size = 4

Figure 2: Test accuracy (%) by using different normalization combinations on ResNet18. Redder represents higher accuracy. Slash means the failure of training. The diagonal cells are the traditional homologous normalization, while other cells are heterologous normalization.

**Ablation study.** We investigate different configure of channel groups for EN on ResNet18. When the mean and the std are computed from the same channel groups, the normalization is homologous. Otherwise, the normalization is heterologous. When the group is $c$, which means the number of channels, EN is equal to BN. As shown in Figure 3, the diagonal cells are the homologous EN, while other cells are heterologous EN. We can find heterologous normalization is generally valid, especially in the case of the small batch size. Moreover, we can find the model achieves better performance by setting the mean's group to $c$ and varying the std's group. For example, by setting the mean's group to $c$ and the std's group to 1, the model gets the best accuracy when the batch size is 128. By setting the mean's group to $c$ and the std's group to 8, the model gets the best accuracy when the batch size is 4. That is to say, there is no need to extend the pixels to calculate the mean. It is enough to extend the pixels to calculate the std. It is verified again that the heterologous combination (BN,EN) works well.

| mean / std | c | 32 | 16 | 8 | 4 | 1 |
|---|---|---|---|---|---|---|
| c | 95.05 | 94.33 | 93.77 | 93.54 | 93.92 | 93.41 |
| 32 | 94.97 | 93.52 | 92.90 | 92.48 | 92.52 | 92.31 |
| 16 | 94.88 | 93.21 | 93.60 | 91.96 | 92.60 | 92.73 |
| 8 | 94.89 | 91.89 | 93.57 | 91.98 | 92.35 | 92.32 |
| 4 | 95.03 | 92.93 | 92.50 | 92.73 | 93.09 | 91.69 |
| 1 | 95.18 | 92.04 | 91.17 | 88.64 | 92.98 | 91.66 |

(a) Batch size = 128

| mean / std | c | 32 | 16 | 8 | 4 | 1 |
|---|---|---|---|---|---|---|
| c | 84.59 | 94.14 | 94.26 | 94.43 | 94.14 | 94.19 |
| 32 | 95.04 | 94.17 | 94.48 | 94.47 | 94.08 | 93.70 |
| 16 | 94.95 | 94.50 | 94.41 | 94.07 | 94.16 | 93.71 |
| 8 | 95.17 | 94.69 | 94.09 | 94.28 | 94.01 | 94.07 |
| 4 | 94.72 | 94.34 | 94.27 | 93.95 | 93.86 | 93.70 |
| 1 | 94.93 | 94.37 | 94.28 | 93.87 | 93.89 | 93.69 |

(b) Batch size = 4

Figure 3: Test accuracy (%) by using different channel groups to calculate Extend Normalization's mean and std. Redder represents higher accuracy. When the group is $c$, which means the number of channels, Extend Normalization is equal to Batch Normalization.

**Comparing to Switchable Normalization (SN).** SN mixes both the mean and the std from BN, IN, and LN. SN switches between them by learning their importance using trainable parameters. The selectivity of normalizers makes SN robust to batch size. We compare SN and HN using ResNet18 with various batch sizes. HN adopts the heterologous combination (BN,EN) where the group of EN is 1. Figure 4 and Table 2 show the results. We can find that HN outperforms SN on all batch sizes except batch size 8. For example, HN's accuracy is 0.50% higher than SN's with large batch size 128 and 0.58% higher with small batch size 2.

**Different networks.** We further evaluate HN on different networks: MobileNetV2_x0.35, MobileNetV2_x1.0, ShuffleNetV2_x1.5, and Densenet121. BN, IN, LN, GN, and SN are homologous. HN adopts the heterologous combination (BN,EN) where the channel group of EN is 1. We also

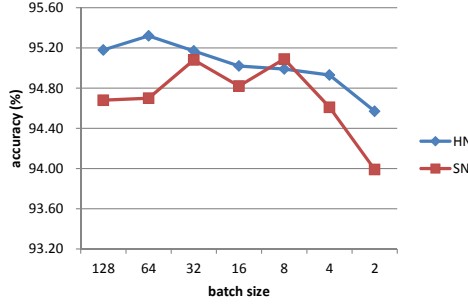

Figure 4: Comparisons between SN and HN with different batch sizes.

Table 2: Test accuracies (%) of SN and HN with different batch sizes. Diff represents HN's accuracy subtracts SN's.

| Batch size | SN | HN | Diff |
|---|---|---|---|
| 128 | 94.68 | 95.18 | 0.50 |
| 64 | 94.70 | 95.32 | 0.62 |
| 32 | 95.08 | 95.17 | 0.09 |
| 16 | 94.82 | 95.02 | 0.20 |
| 8 | 95.09 | 94.99 | -0.10 |
| 4 | 94.61 | 94.93 | 0.32 |
| 2 | 93.99 | 94.57 | 0.58 |

evaluate HN_G8, which refers to heterologous combination (BN,EN_G8) where the channel group of EN is 8 ( A larger group cannot be divided by these network channels since they have fewer channels.) Table 3 shows the test accuracy of these networks with batch size 128. BN achieves much better performance than IN, LN, GN. SN improves the accuracy of MobileNetV2_x1.0, but degrades other networks. On MobileNetV2_x0.35, MobileNetV2_x1.0 and ShuffleNetV2_x1.5, HN_G8 and HN achieve the best and the second best accuracy compared to other normalizations. On DenseNet121, HN_G8 achieves comparable accuracy to BN, whose accuracy is the best.

Table 3: Test accuracies (%) on different networks. Slash means the failure of training.

| | BN | IN | LN | GN | SN | HN | HN_G8 |
|---|---|---|---|---|---|---|---|
| MobileNetV2_x0.35 | 89.60 | 82.96 | / | 80.40 | 88.95 | **89.91** | **90.20** |
| MobileNetV2_x1.0 | 90.92 | 87.42 | 53.48 | 84.93 | 91.40 | **91.49** | **91.92** |
| ShuffleNetV2_x1.5 | 91.40 | 86.83 | 86.75 | 86.99 | 90.41 | **92.64** | **92.81** |
| DenseNet121 | **95.20** | 92.99 | 86.69 | 91.08 | 94.39 | 94.03 | **95.11** |

## 4.2 CIFAR-100

CIFAR-100 is similar to CIFAR-10 but with 100 classes. The experiment settings are the same as Section 4.1. Table 4 shows the test accuracy using ResNet18 on CIFAR-100. With the batch size 128, BN achieves the best performance 77.91%. HN and HN_G32 have close accuracy, surpassing other normalization except BN. HN's accuracy is 5.11%, 8.32%, 3.11%, and 0.90% higher than IN, LN, GN, and SN, respectively. With batch size 4, HN and HN_G32 have close accuracy, surpassing all other normalization. HN_G32's accuracy is 0.55%, 5.45%, 4.69%, 2.67%, 1.57% higher than BN, IN, LN, GN, SN, respectively.

Table 4: Test accuracy (%) on CIFAR-100.

| | BN | IN | LN | GN | SN | HN | HN_G32 |
|---|---|---|---|---|---|---|---|
| Batch size = 128 | **77.91** | 71.65 | 68.44 | 73.65 | 75.86 | **76.76** | 76.60 |
| Batch size = 4 | 76.60 | 71.70 | 72.46 | 74.48 | 75.58 | **77.02** | **77.15** |

## 4.3 CALTECH256

CalTech256 has 30607 images and 256 classes. We use 80% images as the training data and 20% images as the test data. Some images have only one channel, and we copy the one channel to form three channels. Other experiment settings are the same as Section 4.1, except CalTech256's image size is 224x224. CalTech256 has more classes, and each class has fewer training examples. Table 5 shows the test accuracy using ResNet18 on CalTech256. With batch size 128, SN achieves the best performance, and HN_G32 achieves the second best performance. HN_G32's accuracy is 0.75%, 5.61%, 7.44%, and 3.44% higher than BN, IN, LN, and GN, respectively. With batch size 4, HN

and HN_G32 surpass all other normalization. HN_G32's accuracy is 1.64%, 8.52%, 9.01%, 4.66%, 1.55% higher than BN, IN, LN, GN, SN, respectively.

Table 5: Test accuracy (%) on CalTech256.

|  | BN | IN | LN | GN | SN | HN | HN_G32 |
|---|---|---|---|---|---|---|---|
| Batch size = 128 | 56.39 | 51.52 | 49.69 | 53.69 | **58.44** | 56.32 | **57.14** |
| Batch size = 4 | 58.512 | 51.64 | 51.151 | 55.492 | 58.60 | **59.491** | **60.16** |

## 4.4 IMAGENET

ImageNet classification dataset (Russakovsky et al., 2015) has 1.28M training images and 50,000 validation images with 1000 classes. The training images are cropped with random size of 0.08 to 1.0 of the original size and a random aspect ratio of 3/4 to 4/3 of the original aspect ratio, and then resized to 224x224. Then random horizontal flipping is made. The validation image is resized to 256x256, and then cropped by 224x224 at the center. Each channel of the input is normalized into 0 mean and 1 std globally. Weight decay of 0.0001, and SGD with 0.9 momentum are used. On ImageNet, we evaluate ResNet18 with BN, GN, and HN. HN adopts the heterologous combination (BN,EN) where the channel group of EN is 1. Four GPUs are used. We consider batch sizes 256 and 16. The mean and the std of BN and HN are computed within each GPU. For the batch size of 256 (64 images per GPU), the initial learning rate is set to 0.1. And for the batch size of 16 (4 images per GPU), the initial learning rate is set to $0.1 * 16/256$, following the linear scaling rule. We train the network for 100 epochs and decrease the learning rate by 10x at 30, 60, and 90 epochs.

The results of ImageNet are shown in Figure 5 and Table 6. From Figure 5(a), we can see HN achieves close performance to BN, better than GN, with large batch size. From Figure 5(b), we can see HN achieves close performance to GN, better than BN, with small batch size. As shown in Table 6, with 64 images per GPU, BN achieves the best validation accuracy of 70.37%. HN gets 70.12%, worse than BN by 0.25%, but better than GN by 1.35%. With 4 images per GPU, the accuracy of BN decreases to 65.78%. GN achieves the best accuracy of 69.08%. HN achieves 68.54%, worse than GN by 0.54%, better than BN by 2.76%.

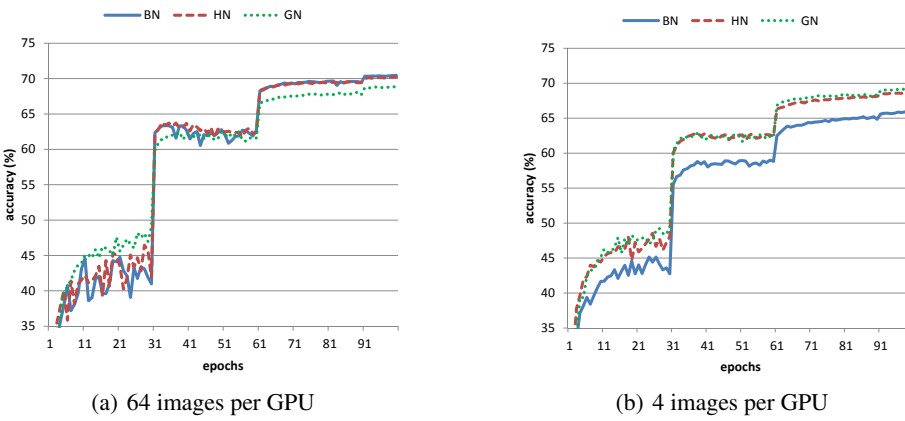

(a) 64 images per GPU      (b) 4 images per GPU

Figure 5: Test accuracy of ResNet18 on ImageNet vs. the number of training epoch.

## 4.5 COMBINATION WITH SWITCHABLE NORMALIZATION.

Though we have shown HN outperforms SN in most situations, we should know HN and SN are not mutually exclusive. We can combine HN and SN together to improve the performance. We evaluate two strategies for combining HN and SN. One is (BN,BN+EN), where the mean is calculated like BN, and the std is the combination of BN and EN. Another is (BN,BN+IN+LN), where the mean is calculated like BN, and the std is the combination of BN, IN, and LN, as the same as SN. Table 7 shows ResNet18 accuracies on CalTech256 with batch size 128. We can find that combinations of

HN and SN further improve accuracies with large and small batch sizes. (BN,BN+EN) outperforms both (BN,BN) and (BN,EN). Since the moving average value is used, there is no need to compute the mean and the std at inference time for (BN,BN+EN). (BN,BN+IN+LN) achieves the best accuracy, but the std needs to be calculated at inference time since it includes IN and LN.

Table 6: Test accuracy (%) of ResNet18 on ImageNet.

|  | BN | GN | HN |
| --- | --- | --- | --- |
| 64 images per GPU | 70.37 | 68.77 | 70.12 |
| 4 images per GPU | 65.78 | 69.08 | 68.54 |

Table 7: Test accuracy (%) on Cal-Tech256 when combining Heterologous Normalization with Switchable Normalization.

| mean | std | Batch size | |
| --- | --- | --- | --- |
|  |  | 128 | 4 |
| BN | BN | 56.39 | 58.51 |
| BN | EN | 56.32 | 59.49 |
| BN, LN, IN | BN, LN, IN | 58.44 | 58.60 |
| BN | BN, EN | 57.95 | 59.81 |
| BN | BN, LN, IN | **58.65** | **61.15** |

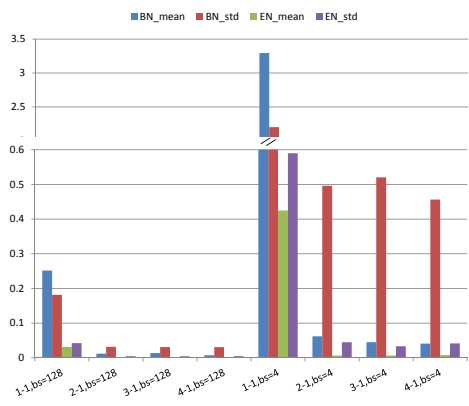

Figure 6: Comparion of statistics' std which reflects the statistics' fluctuation quantitatively. Neuron i-j represents the $j_{th}$ neuron of the $i_{th}$ layer, and bs refers to batch size.

## 4.6 ANALYSIS

We train a four-layer fully-connected (FC) network on MNIST. Each FC layer have 128 neurons, BN and ReLU are used after the FC layer. We train the network with batch sizes 128 and 4. The learning rate is 0.1 when the batch size is 128. It is set to $0.1 * 4/128$ when the batch size is 4, following the linear scaling rule. We train the network for 20 epochs. For each layer, we calculated each neuron's mean and std along the batch dimension (BN_mean and BN_std). Moreover, we calculated the mean and the std of the universal neurons (EN_mean and EN_std). We use i-j to represent the $j_{th}$ neuron of the $i_{th}$ layer. Figure 7 shows the statistics evolutions of different neurons over the last 200 training steps. To quantify the fluctuation, we show the std of these statistics themselves in Figure 6. We have the following observations:

- Statistics of BN are more fluctuant than EN. That is to say, the noises brought by BN are more than EN since EN extends the pixel set for estimating statistics. When the batch size is 128, the noises brought by BN are moderate thus are helpful for training.

- The difference between BN_mean and EN_mean is larger than the difference between BN_std and EN_std. This phenomenon is obvious in Figure 7(a), 7(e), 7(g). That explains why replacing BN_std with EN_std is more effective than replacing BN_mean with EN_mean.

- BN_std have more fluctuations, while BN_mean is flatter than BN_std. When the batch size is 4, as shown in Figure 6, 7(d), 7(f), 7(h), BN_std's fluctuations become dramatic. That implies the small batch size problem of BN is caused mainly by the fluctuation of BN_std. EN_std has smaller fluctuation, thus using EN_tsd instead of BN_std can alleviate the small batch size problem of BN.

- Statistics in the first layer have more significant fluctuations than following layers, as shown in Figure 6, 7(a) and 7(b), especially in the case of small batch size. The lower layer is closer to the data layer, thus is influenced more easily by the noise in the data. After passing through multiple learnable layers, the noises diminish comparatively. That suggests different layers could use different strategies to control the noise. For example, We may extend the pixel set for calculating both the mean and the std in the lower layer. Moreover, the extension degree (the group in EN) could be different for different layers.

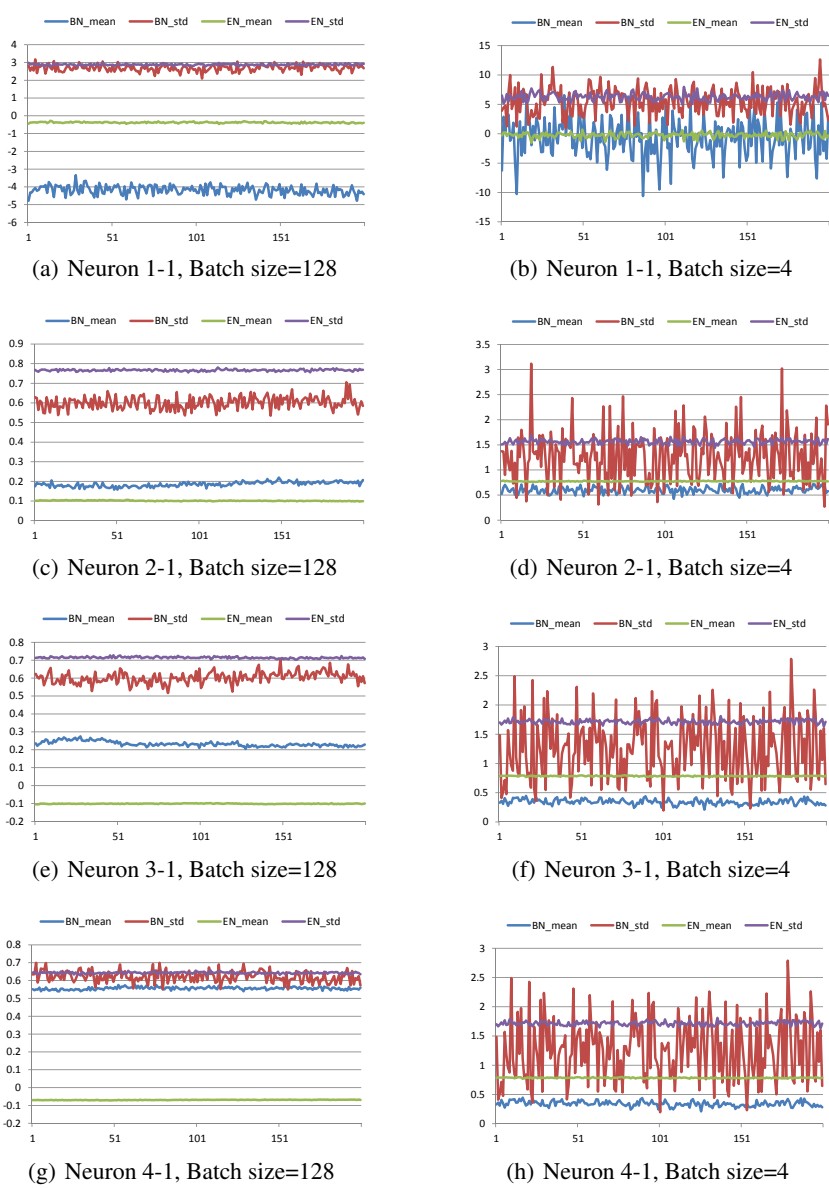

Figure 7: Statistics evolution over the course of training. Neuron i-j represents the $j_{th}$ neuron of the $i_{th}$ layer. Horizontal axis represents training step. The small batch size problem of Batch Normalization is caused mainly by the fluctuation of the std.

## 5 CONCLUSION

In this paper, we propose Heterologous Normalization (HN), which computes the mean and the std from different pixel sets to take advantage of different normalizations. HN is a general method that can use different strategies to compute the mean and the std. Moreover, we propose a particular combination that works well in most situations: calculating the mean along the (N, H, W) dimensions as the same as BN, while calculating the std along the (N,C,H,W) dimensions. It not only maintains the advantage of Batch Normalization but also alleviates the problem caused by the small batch size. HN surpasses or achieves comparable performance than existing homologous methods with large or small batch sizes on various datasets. Besides, HN can be combined with BN together to improve the performance further. Moreover, we explain why HN conquered the small batch size problem by analyzing the evolution of the statistics over the course of training.

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
