# OpenReview forum: "Heterologous  Normalization"
_ICLR.cc/2022/Conference — ICLR 2022 Submitted_

### Official Review · Reviewer_Zi9s · 2021-11-02

**Correctness:** 3
**Technical Novelty And Significance:** 3
**Empirical Novelty And Significance:** 2
**Recommendation:** 5
**Confidence:** 4

**Main Review:**

This paper is easy to follow and the method seems somewhat novel.

However, there are several weaknesses that lead me to give reject, not good enough in this stage.

1) Although this method seems somewhat novel, the experiments do not support it enough. Specifically,  the results should be run at least 5 times, reporting it mean and std for CIFAR and CALTECH256. Besides, the results report for GN in CIFAR is inferior to BN, which is unreasonable to me. The author should check their parameters setting or use some official report results for double-checking.

2) This paper lacks sufficient theory to explain why the combination strategy works well in most situations: calculating the mean along the (N, H, W) dimensions as the same as BN, while calculating the std along the (N,C,H,W) dimensions.

**Summary Of The Paper:**

This paper proposes a new technique called Heterologous Normalization to help train modern deep networks. The main method is easy to follow. In contrast to conventional homologous normalization methods(e.g BN, IN), HN computes normalization’s mean and standard deviation from different pixel sets. The experiments show HN is slightly better than BN, and GN in some cases.






**Summary Of The Review:**

Given the current weakness of experiments and theory part, I tend to weak reject this stage.

---

### Official Review · Reviewer_5kEH · 2021-11-02

**Correctness:** 3
**Technical Novelty And Significance:** 3
**Empirical Novelty And Significance:** 2
**Recommendation:** 5
**Confidence:** 4

**Main Review:**

The presented work is novel and well embedded
into other works on normalization methods. Overall, it is a
variation of an existing, but highly important, method.
The relevance of the work is somewhat limited since the
reported performance improvements are relatively far from
current state-of-the-art. The work appears technically sound.
All reported performance values lack error bars.
A hypothesis and/or explanation of why heterologous estimation
of normalization parameters work is missing.

### Strengths

a) The work is clearly written, well structured and well explained.
The structure and notation of the paper follow standards in the field.


b) The aim of the paper is clearly written and the work is well embedded
into related works. Differences and similarities are explained and
how other normalization methods arises as special cases of heterologous
normalization.

c) The authors perform experiments on relevant benchmarking datasets and
also perform and ablation study that is well-motivated.


### Weaknesses

a) The presented work in the current form is limited to images with
their specific data structure and layout, i.e. a sample dimension,
a channel, a width, and a height dimension. However, normalization methods
play a role in almost all data types and architectures (graph neural networks,
recurrent neural networks, transformers). The authors should at least
provide a perspective of the relevance and applicability of their method
to other data types and/or a more general formulation of their approach.


b) All experiments and reported performance values are presented without
repetitions and thus without error bars. Thus, difference in predictive
performance could arise just by chance. The authors should provide error
bars for all experiments to demonstrate that their reported performance
improvements do not just arise by chance.

c) The experiments are performaned within a setting that is very far
from current state-of-the-art at predictive performance. E.g. on ImageNet,
the authors architectures operate at an accuracy of 65-70% (Table 6
and Figure 5), whereas VGG was reported at 75% in 2014 and current
architectures such as NFNets [1] operate close to 90%. Therefore, the relevance
of the proposed approach is relatively low. The authors should perform
experiments with current state-of-the-art models to assess whether
their approach is relevant in those architectures and experiments.

d) Figures 4,5,6,7 could be improved to follow scientific standards.

e) There is a lack of a hypothesis or at least explanation why this strategy
to estimate mean and standard deviation based on different sample sets
improves batch normalization aside from the trivial fact that a larger
number of samples are used. In that regard, also other strategies,
such as using geometric averages of past parameter estimates
should be discussed.

**Summary Of The Paper:**

The authors present a strategy to calculate the normalizing parameters of batch normalization, i.e. mean and standard deviation, based on different sets of samples.

**Summary Of The Review:**

In the current form, I consider this work to be below the bar to
be accepted due to the problems and limitations mentioned above.
I am willing to increase my score if the mentioned improvements
are performed and presented in the paper.

---

### Official Review · Reviewer_cQjf · 2021-11-02

**Correctness:** 3
**Technical Novelty And Significance:** 2
**Empirical Novelty And Significance:** 2
**Recommendation:** 5
**Confidence:** 4

**Main Review:**


The proposed strategy is quite simple. All underlying computations are built on previous methods. It seems that there is not much to argue about on the implementation and realizability of the proposed strategy. Therefore, the main focus of the review would be more on the validity of the design principle and on the empirical evaluation and analysis:

1. As the statistic of standard deviation measures how widely the values are spreading over around the mean of the set, it is odd that the mean $\mu_{S_i}$ used in the normalization is different from the mean $\mu_{S_i\'}$ used for estimating the standard deviation $\sigma_{S_i\'}$. That is, the normalization formula actually includes two different *mean*s. It is not easy to rationalize what the formula actually calibrates.
2. It is unclear why heterologous combinations $(BN, *)$ shown in Fig. 2 all achieve good performances on batch size 128 and what the results might imply. There are no further discussions on this point.
3. The claim in the last paragraph of the introduction is also ambiguous and not well justified.
What does it mean by "*equilibrium between generalization and stability*"? Is "*controlling the **number** of pixels to calculate the statistics*" really the key?
4. It is reasonable that "the difference between BN mean and EN mean is larger than the difference between BN std and EN std." However, it does not seem to be very meaningful to draw a conclusion as "*that explains replacing BN std with EN std is more effective than replacing BN mean with EN mean*". Having similar values does not justify the validity and effectiveness of replacing BN std with EN std.
Furthermore, is the fluctuation in the standard deviation indeed the main issue for normalization with a small batch size? If that is the case, then what is the appropriate range for the standard deviation? the smaller the better?




**Summary Of The Paper:**

This paper presents a heterologous normalization method that estimates the mean and the variance from different pixel sets for training deep networks. It is claimed in the paper that this kind of mixed statistics for normalization provides better and more stable performance on learning a deep model. Several experiments of classification tasks on CIFAR-10, CIFAR-100, Caltech-256, and ImageNet datasets are shown to support the claim. The paper also presents an analysis to illustrate the fluctuation in the statistics during training.

**Summary Of The Review:**

The idea of this paper is simple, and the proposed method is straightforward and seems very easy to implement: Given all available options of existing normalization techniques, one may handily combine different techniques for estimating the mean and the variance. In practice, there might not be apparent drawbacks of trying such a heuristic strategy. On the other hand, the strategy still lacks more solid technical supports and insights to justify why it works and how to apply it for different scenarios. It seems that, for most cases, it would be much easier for the user to adopt a simpler strategy like "*if memory is sufficient, use a large batch size + batch normalization; otherwise, use a small batch size + group normalization*". The experimental results look fair, but the improvements are not very significant. Therefore, whereas this paper presents a new strategy to carry out normalization for training deep networks, its contributions are marginally significant and require further investigations.

---

### Official Review · Reviewer_fNu6 · 2021-11-08

**Correctness:** 3
**Technical Novelty And Significance:** 2
**Empirical Novelty And Significance:** 2
**Recommendation:** 5
**Confidence:** 4

**Main Review:**

** Pros

1)	The proposed method is very simple and easy to implement. I particularly like the insights of using different sets to derive the statistics, which implies the mean and standard deviation terms in the normalization layer may work in different manners. It could be a good start to understand how normalization works in modern neural networks.
2)	When batch size varies, HN behaves more robust than the counterpart BN. Unlike instance-level normalizations such as IN, GN which require to calculate the statistics in inference time, HN can be totally cost-free (e.g. the configuration of BN+EN) for inference.

** Cons

1)	Unfortunately, the proposed HN seems not to beat the state-of-the-art methods especially on large datasets like ImageNet. According to Table 6, HN is slightly worse than BN for large batch size, and worse than GN for small batch size. To stabilize the training on small batch sizes, a widely-used recipe GN+WS [*1] is mentioned in many pervious methods, but not compared in this paper. Moreover, as shown in Fig 4, it seems HN still suffers from performance drop as batch size decreases. Although it is better than the counterpart BN, a recent work MABN [*2] is able to eliminate the degradation even when the batch size is 1, which is also cost-free in inference time. So, I think HN may not be an optimal choice for practice use.
2)	In Sec 4.6, to understand how HN works, that paper points that the std term of BN has more fluctuations than the mean term under small batch sizes. However, I find that in all cases EN introduces more stable statistics than BN, but according to Fig 2, EN+EN configuration performs not as good as EN+BN, which I think cannot be attributed to the statistic noise. I recommend the authors may investigate more in-depth interpretations. For example, does EN+BN recipe benefit from the implicit regularization effect of BN?

[*1] Qiao, Siyuan, et al. "Weight standardization." (2019).

[*2] Yan, Junjie, et al. "Towards stabilizing batch statistics in backward propagation of batch normalization." ICLR 2020.


**Summary Of The Paper:**

The paper introduces Heterologous Normalization (HN), an alternative to normalization techniques in neural networks such as BN, LN, etc. The key insight is, the optimal statistics of mean and standard deviation may be derived from different pixel sets respectively. Based on the observation, the proposed HN calculates the mean in BN’s style, while following the way of EN to stabilizing the variance. Experiments shows HN achieves comparable or slightly better performances than BN’s in normal batch sizes. While for very small batch sizes (e.g. 4), HN significantly outperforms BN by a large margin.

**Summary Of The Review:**

Although the paper introduces some interesting insights, due to many similar works in this direction, I do not think the contribution of the paper is significant.

---

### Decision · Program_Chairs · 2022-01-20

**Decision:**

Reject

**Comment:**

All reviewers recommended reject. No responses from the authors.